# Effectiveness of a Learning Pathway on Food and Nutrition in Amyotrophic Lateral Sclerosis

**DOI:** 10.3390/nu17152562

**Published:** 2025-08-06

**Authors:** Karla Mônica Dantas Coutinho, Humberto Rabelo, Felipe Fernandes, Karilany Dantas Coutinho, Ricardo Alexsandro de Medeiros Valentim, Aline de Pinho Dias, Janaína Luana Rodrigues da Silva Valentim, Natalia Araújo do Nascimento Batista, Manoel Honorio Romão, Priscila Sanara da Cunha, Aliete Cunha-Oliveira, Susana Henriques, Luciana Protásio de Melo, Sancha Helena de Lima Vale, Lucia Leite-Lais, Kenio Costa de Lima

**Affiliations:** 1Health Sciences Graduate Program, Federal University of Rio Grande do Norte, Natal 59012-970, Brazil; karla.coutinho@lais.huol.ufrn.br (K.M.D.C.); sancha.vale@ufrn.br (S.H.d.L.V.); limke@uol.com.br (K.C.d.L.); 2Laboratory for Technological Innovation in Health (LAIS), Federal University of Rio Grande do Norte, Natal 59012-300, Brazil; hrabeloufrn@gmail.com (H.R.); felipe.ricardo@lais.huol.ufrn.br (F.F.); karilany@lais.huol.ufrn.br (K.D.C.); ricardo.valentim@lais.huol.ufrn.br (R.A.d.M.V.); aline.dias@lais.huol.ufrn.br (A.d.P.D.); janaina.lrsv@lais.huol.ufrn.br (J.L.R.d.S.V.); natalia.araujo@lais.huol.ufrn.br (N.A.d.N.B.); manoel.romao@lais.huol.ufrn.br (M.H.R.); priscila.sanara@lais.huol.ufrn.br (P.S.d.C.); luciana.protasio@lais.huol.ufrn.br (L.P.d.M.); 3Center for Interdisciplinary Studies (CEIS20), University of Coimbra, 3000-457 Coimbra, Portugal; alietecunha@esenfc.pt; 4Health Sciences Research Unit: Nursing (UICISA: E), Nursing School of Coimbra (ESEnfC), 3046-851 Coimbra, Portugal; 5Department of Biomedical Engineering, Federal University of Rio Grande do Norte, Natal 59078-900, Brazil; 6Health Innovation and Management Graduate Program, Federal University of Rio Grande do Norte, Natal 59012-300, Brazil; 7Center for Global Studies (CEG), Universidade Aberta, 1269-001 Lisbon, Portugal; susana.henriques@uab.pt; 8Department of Nutrition, Federal University of Rio Grande do Norte, Natal 59078-970, Brazil

**Keywords:** motor neuron disease, nutrition, health education, online learning

## Abstract

**Background/Objectives:** Health education plays a vital role in training health professionals and caregivers, supporting both prevention and the promotion of self-care. In this context, technology serves as a valuable ally by enabling continuous and flexible learning. Among the various domains of health education, nutrition stands out as a key element in the management of Amyotrophic Lateral Sclerosis (ALS), helping to prevent malnutrition and enhance patient well-being. Accordingly, this study aimed to evaluate the effectiveness of the teaching and learning processes within a learning pathway focused on food and nutrition in the context of ALS. **Methods:** This study adopted a longitudinal, quantitative design. The learning pathway, titled “Food and Nutrition in ALS,” consisted of four self-paced and self-instructional Massive Open Online Courses (MOOCs), offered through the Virtual Learning Environment of the Brazilian Health System (AVASUS). Participants included health professionals, caregivers, and patients from all five regions of Brazil. Participants had the autonomy to complete the courses in any order, with no prerequisites for enrollment. **Results:** Out of 14,263 participants enrolled nationwide, 182 were included in this study after signing the Informed Consent Form. Of these, 142 (78%) completed at least one course and participated in the educational intervention. A significant increase in knowledge was observed, with mean pre-test scores rising from 7.3 (SD = 1.8) to 9.6 (SD = 0.9) on the post-test across all courses (*p* < 0.001). **Conclusions:** The self-instructional, technology-mediated continuing education model proved effective in improving participants’ knowledge about nutrition in ALS. Future studies should explore knowledge retention, behavior change, and the impact of such interventions on clinical outcomes, especially in multidisciplinary care settings.

## 1. Introduction

Amyotrophic lateral sclerosis (ALS) is a rare, progressive neurodegenerative disease that affects motor neurons, leading to muscle atrophy, respiratory dysfunction, and eventual loss of autonomy. It is a complex condition with multifactorial origins, including genetic, environmental, and lifestyle factors [1,2,3,4]. One of the most critical clinical challenges in ALS is nutritional deterioration, which contributes to poor quality of life, accelerated disease progression, and increased mortality. Studies estimate that 16 to 55% of ALS patients are already malnourished at diagnosis, and the risk increases as the disease advances [5,6,7]. In this context, maintaining adequate nutritional status is recognized as a key prognostic factor, requiring early and ongoing interventions that involve patients, caregivers, and healthcare professionals in coordinated care [7,8,9].

Health education plays a vital role in strengthening health systems by promoting continuous learning, contextualized practices, and interprofessional collaboration [10,11,12]. In ALS, education is especially important to address the evolving care needs and the complexity of disease management, particularly in the domain of nutrition [13]. Empowering caregivers—both formal and informal—and health professionals with evidence-based knowledge can support self-care, shared decision-making, and better patient outcomes [14,15]. However, many caregivers and professionals lack adequate training in ALS-specific nutritional care, especially in low-resource settings or where health services are fragmented [16,17].

To expand access to training, technology-mediated health education has emerged as a promising tool. Virtual Learning Environments (VLEs) and Massive Open Online Courses (MOOCs) allow for flexible, large-scale learning opportunities and the dissemination of up-to-date health content [18,19,20]. In Brazil, the AVASUS platform (Virtual Learning Environment of the Brazilian Health System) offers free, open-access courses developed in partnership with the Ministry of Health and the Laboratory for Technological Innovation in Health (LAIS/UFRN) [21,22]. With over 2.5 million certificates issued, AVASUS plays a strategic role in national capacity building. Nevertheless, like other MOOCs, this model is not without limitations. Barriers such as digital inequality (especially between rural and urban regions), low completion rates, and difficulties in sustaining learner engagement—particularly among caregivers with no formal healthcare background—must be critically acknowledged [22,23]. While MOOCs offer scalability and flexibility, their effectiveness depends on the relevance of content, the support offered to learners, and the degree to which they are tailored to diverse user profiles [24,25,26,27].

To address these challenges in ALS care, the learning pathway “Food and Nutrition in ALS” was developed and implemented on the AVASUS platform. It comprises four self-instructional MOOCs designed to educate healthcare professionals, caregivers, and patients about the role of nutrition in ALS management. Given the centrality of nutritional care in the clinical trajectory of ALS, this study aimed to evaluate the effectiveness of the teaching–learning process within this educational pathway.

## 2. Materials and Methods

### 2.1. Ethical and Legal Aspects

The study was approved by the Research Ethics Committee of the Onofre Lopes University Hospital (CEP/HUOL) of the Federal University of Rio Grande do Norte (UFRN), under No. 4.431.327, on 1 December 2020. Participation was voluntary and contingent upon signing an Informed Consent Form (ICF). The data extracted from the digital platform (AVASUS) followed the standards of the Brazilian General Data Protection Law. All personally identifiable information—such as user IDs and geolocation data—was removed prior to analysis. The final dataset, available on the public repository Zenodo (https://doi.org/10.5281/zenodo.15678383 accessed on 21 July 2025), contains only anonymized information and complies with institutional ethical standards.

### 2.2. Study Design and Population

This was a longitudinal, experimental before-and-after study with a quantitative approach, conducted between June 2021 and May 2024. The target population included health professionals, formal and informal caregivers, and individuals diagnosed with Amyotrophic Lateral Sclerosis (ALS), of both sexes, from across Brazil.

Participants were recruited through invitations disseminated in collaboration with the Brazilian Association of Amyotrophic Lateral Sclerosis (AbrELA), its regional networks (ARELA-RS and ARELA-MG), and specialized ALS care centers. The invitation was also shared via digital platforms and social media groups that support the ALS community, including Instagram, Facebook, and WhatsApp.

Inclusion criteria for participation in the study were: (i) completion of at least one course from the “Food and Nutrition in ALS” learning pathway, and (ii) completion of both assessments—pre-test and post-test—available on the platform. Individuals under 18 years of age were excluded. Additionally, participants who dropped out or failed to complete the assessments were considered sample losses. A flowchart detailing the process of participant recruitment, consent, course engagement, and exclusion is shown in Figure 1.

### 2.3. Educational Intervention

The educational intervention consisted of a learning pathway titled “Food and Nutrition in ALS”, composed of four asynchronous, self-instructional MOOCs hosted on AVASUS. These courses were developed by a team of nutritionists with expertise in the field. The pathway includes the following courses: (i) Food and Nutrition for ALS Patients, (ii) Dietary Modifications for ALS Patients, (iii) Tube Feeding for ALS Patients, and (iv) Specific Nutritional Considerations for ALS Patients. The total workload for the complete pathway is 65 h. These courses cover essential topics such as nutritional care in ALS, dysphagia management, nutritional assessment, and enteral nutrition. Each course was designed to be completed independently and flexibly, with no prerequisites or fixed sequence, allowing participants to progress at their own pace. Before and after each course, participants were required to complete a pre-test and a post-test, each consisting of six multiple-choice questions related to the course content. The data report on this learning pathway was recently published [28].

### 2.4. Evaluation of Learning Outcomes

To evaluate the effectiveness of the teaching and learning process, participants’ knowledge acquisition was assessed using structured, objective multiple-choice tests administered before and after each course. These instruments were specifically developed for this study by a team of nutritionists with expertise in ALS and instructional design. Each course in the learning pathway included a pre-test consisting of 10 randomized multiple-choice questions aligned with the course content, which was mandatory for enrollment. At the end of each course, participants completed a post-test composed of 10 questions covering the same topics, allowing for direct comparison of knowledge acquisition.

Each correct answer was worth one point, with a maximum possible score of 10. Although no formal cutoff was established to define “success,” an improvement in the post-test score compared to the pre-test was used as an indicator of learning. These instruments were not previously validated but were reviewed by subject matter experts to ensure content accuracy, relevance, and clarity. Only participants who completed both the pre-test and post-test for at least one course were included in the final analysis.

### 2.5. Data Collection

Descriptive data were collected through an anonymized, structured questionnaire administered at the time of study enrollment. The questionnaire included both multiple-choice questions and open-ended questions that required written responses. It addressed participants’ personal information, self-reported occupation, and prior experience with ALS care and technology-mediated education. Key variables assessed included gender, age, geographic region in Brazil, educational level, occupation, prior experience with online courses, experience caring for individuals with ALS, and self-reported baseline knowledge regarding nutrition in ALS.

Data related to the intervention—specifically, course participation, pre- and post-test completion, and performance—were extracted directly from the AVASUS platform. All data were fully anonymized to ensure participant privacy. Unique identifiers, geolocation data, and any personally identifiable information were removed prior to analysis.

### 2.6. Data Processing

Following data acquisition, a structured workflow was established to integrate, transform, and organize the information for analytical purposes and subsequent dissemination of findings to the scientific community. Data processing was conducted in a computing environment configured with Python version 3.10.12, utilizing auxiliary libraries such as NumPy (v2.0.2), Pandas (v2.2.2), Matplotlib (v3.10.0), and Seaborn (v0.13.2). The workflow was divided into three main stages: (i) data integration and standardization, (ii) feature extraction, and (iii) feature selection.

### 2.7. Data Analysis

Data from the learning pathway on “Food and Nutrition in ALS” were analyzed using descriptive statistics, allowing for the exploration and summary of key properties and characteristics of the dataset.

## 3. Results

Of the 14,263 individuals enrolled in the pathway, 182 consented to participate in the study, and 142 completed at least one course and both tests. These participants were therefore included in the final analysis. Most participants were from the Southeast and Northeast regions of Brazil (Figure 2).

### 3.1. Sociodemographic and Professional Characteristics of Participants

Regarding gender, the majority of participants were female, accounting for approximately 86% of the sample (Figure 3A). The ages of the 182 participants ranged from 19 to 64 years, with a mean age of 40 years. In terms of educational level, Figure 3B, 48% of the participants had completed a higher education degree, while 26% had incomplete higher education. Figure 3C presents the five most frequently reported occupations, with nutritionists comprising the largest group (33%). Although the target population included ALS patients, formal and informal caregivers, and health professionals, only caregivers and health professionals completed the full requirements for inclusion in the final analysis. No ALS patients completed both the pre- and post-tests necessary for study participation. Among the 142 participants included in the final analysis, 47% identified as healthcare professionals, 26% as caregivers, and 27% as informal or family caregivers and/or patients.

Participants were also assessed regarding their previous experience with technology-mediated courses, their involvement in ALS patient care, and their knowledge of nutrition in the context of ALS. The results showed that over 80% of participants had already taken technology-mediated courses. Additionally, 25% of the participants reported having experience in caring for ALS patients. Among these, more than 57% indicated some prior knowledge of ALS-related nutrition. The main sources of this knowledge included online courses (21%), in-person courses (8%), and other avenues such as social networks, scientific publications, and academic events.

### 3.2. Educational Intervention of the Learning Pathway

Among the 182 participants surveyed, 142 (78%) completed at least one course from the learning pathway “Food and Nutrition in ALS”. Accordingly, two analytical approaches were employed to evaluate the intervention. The first approach assessed each course individually by comparing participants’ pre- and post-test scores. The second approach analyzed the combined results of all courses in the pathway.

Table 1 presents the data from the first approach. The results indicate that the number of courses completed by each participant did not significantly influence their scores in the pathway courses, as determined by the Kruskal–Wallis test.

For course 1 (Food and Nutrition for ALS Patients), the median number of completed modules was 2.00 (interquartile range: 1.00–4.00), with *p*-values of 0.436 for the pre-test and 0.423 for the post-test. For course 2 (Dietary Modifications for ALS Patients), the median number of completed modules was 3.00 (2.00–4.00), and the corresponding *p*-values were 0.255 (pre-test) and 0.959 (post-test). For course 3 (Tube Feeding for ALS Patients), the median number of completed modules was 2.00 (1.00–4.00), with *p*-values of 0.455 (pre-test) and 0.474 (post-test). Finally, for course 4 (Specific Nutritional Considerations for ALS Patients), the median number of completed modules was 3.00 (1.00–3.00), with *p*-values of 0.985 (pre-test) and 0.790 (post-test).

Although no significant association was observed between the number of courses completed and the pre- and post-test scores, the distribution of course completion varied across participants. The highest proportion of participants who completed course 3 had taken only that course. The majority of those enrolled in courses 2 and 4 completed all four courses. For course 1, the distribution was more balanced, with equal proportions of participants completing only one course or all four.

Overall, 35.2% of the participants completed all four courses, 32.2% completed only one course, 16.5% completed two courses, and 16.1% completed three courses. When examining each course individually, course 1 was the most frequently completed, with 28.4% of participants, followed by course 3 (27.6%). Courses 2 and 4 were completed by 21.5% and 22.6% of participants, respectively.

In the overall analysis, the number of courses completed was not significantly associated with pre-test (*p* = 0.594) and post-test scores (*p* = 0.924). However, when analyzing each course separately, significant differences were observed between pre- and post-test scores (Table 1). Courses 1 and 2 showed no significant differences between pre- and post-test scores and formed a higher-performing group. In contrast, courses 3 and 4, which had lower pre- and post-test scores, differed significantly from courses 1 and 2.

## 4. Discussion

The findings of this study highlight the relevance of the learning pathway “Food and Nutrition in ALS”, delivered through the AVASUS. Comprising four self-instructional courses, the learning pathway reached participants across all five regions of Brazil, representing diverse health fields.

Although 14,263 users accessed the learning pathway, only a small proportion met the inclusion criteria, resulting in 182 consenting participants and 142 completing at least one course and both tests. This low conversion rate is consistent with well-documented patterns in open-access MOOCs, which typically experience high attrition due to factors such as voluntary enrollment, lack of formal incentives, time constraints, and the need to complete multiple procedural steps. These challenges are particularly evident in research-linked MOOCs. In a systematic review on participant behavior in health-related MOOCs, authors emphasized that while thousands may register, only a small fraction complete the learning activities or respond to research invitations, underscoring the difficulty of engagement and retention in digital education environments [29]. Our findings mirror this trend and reinforce the need for strategies to improve user engagement in open online learning and research participation.

The geographic distribution of participants, illustrated in Figure 2, demonstrates that individuals from all five macro-regions of Brazil engaged with the learning pathway, despite the small sample size. This national reach reinforces the potential of technology-mediated education—particularly MOOCs delivered through public platforms like AVASUS—to overcome geographic barriers and promote access to continuing education across diverse and often underserved areas. At the same time, the low number of participants in certain regions highlights ongoing challenges in digital inclusion and underscores the need for targeted strategies to increase outreach and participation in remote or resource-limited areas.

Previous research demonstrated that the MOOCs offered through AVASUS democratize access to education, facilitate knowledge exchange at both national and global levels, and provide training and professional development opportunities tailored to diverse socio-cultural contexts [30,31]. Embracing the principles of interculturality, AVASUS incorporates a range of accessibility features, including videos, texts, visual descriptions, audio descriptions, voice-command interfaces, and biometric alternatives. This technological mediation not only streamlines communication but also enriches the learning experience by promoting social inclusion [32,33,34].

The characterization of pathway participants revealed a predominance of females (86%), most of whom had either completed or were pursuing higher education, with nutritionists representing the largest professional subgroup. These findings are consistent with a previous study, which reported two key observations: (i) higher female participation in the learning pathway on food and nutrition in ALS, and (ii) a high prevalence of nutritionists compared to other health professionals participating in the pathway’s courses across Brazil and internationally [28]. Although ALS generally has a higher incidence in men, women are often overrepresented in health-related educational interventions—particularly in roles such as caregivers and healthcare professionals, which comprised a significant portion of our sample. This gender predominance may be partially explained by global patterns in caregiving roles, where women disproportionately assume responsibility for care. Studies indicate that 57% to 81% of all caregivers of the elderly are women, with most care provided at home by family members, especially in chronic and disabling conditions [35].

Informal caregivers, who represented a significant portion of the sample, play a crucial role in the daily management of ALS. However, their educational needs and digital literacy may differ from those of health professionals, potentially influencing course completion and knowledge acquisition. Tailoring educational resources to better meet the needs of this group—including simplified content, family-centered approaches, and emotional support—may enhance their engagement and improve caregiving outcomes. Although ALS patients were included in the target population, none completed the requirements for final inclusion in the study, likely due to the cognitive, motor, or technological limitations imposed by disease progression.

Health professionals comprised the largest subgroup among participants who completed the learning pathway. Notably, 57% of participants reported having satisfactory knowledge of nutrition in ALS, primarily acquired through online (21%) and face-to-face (8%) courses, as reflected in the average pre-test scores (Table 1). This scenario reinforces the relevance of educational initiatives in disseminating specialized knowledge, particularly through virtual learning environments. The high proportion of participants with higher education may have positively influenced engagement and comprehension of the course content, which is supported by the relatively high average pre-test scores observed. This is consistent with the literature, which indicates that educational attainment is associated with better performance in self-directed digital learning environments. Nonetheless, this also underscores the challenge of making MOOCs accessible and effective for individuals with lower educational levels or digital literacy, which should be considered in the design of future interventions.

The analysis of the educational interventions revealed that, although there was no significant association between the number of courses completed and participants’ scores, there was notable variation in course selection. Overall, courses 1 and 3 had the highest participation rates, with 28.4% and 27.6% of the participants, respectively. Courses 1 and 2 were associated with higher average scores compared to courses 3 and 4. This suggests that the content of courses 1 and 2 may have been more accessible or better aligned with the needs and expectations of the target audience.

Participants’ baseline knowledge, as reflected in pre-test scores, and their motivation to engage in the courses likely influenced learning outcomes. Prior familiarity with ALS-related topics may have facilitated content assimilation, especially among professionals already working in the field. Although course quality was not directly assessed, all materials underwent peer review and were developed by experts in clinical nutrition and ALS care. Future studies could include course evaluation metrics and user satisfaction assessments to further understand the relationship between course quality and learning effectiveness.

The variation in the number of participants across the four courses likely reflects the flexible and self-directed structure of the learning pathway, in which participants were free to select only the courses that aligned with their specific interests or needs. This pattern is consistent with findings from other MOOC-based interventions, where optional participation and the absence of external incentives often contribute to uneven engagement levels across modules [29,36].

The use of MOOCs in health education has shown promising potential for expanding access to training opportunities, especially in contexts with limited educational resources. These courses offer flexible, scalable, and often cost-effective ways to reach diverse audiences, including healthcare professionals and caregivers. While previous research supports the value of MOOCs in fostering knowledge dissemination and professional development [22], it is important to acknowledge ongoing challenges related to learner engagement, course completion rates, and the evaluation of learning outcomes.

In the field of public health, digital education strategies have the potential to improve service delivery through health-promoting practices, preventive actions, and increased autonomy of both professionals and users [14,32,37]. In this study, the use of technology-mediated education—featuring open educational resources presented through familiar media formats such as videos and podcasts—contributed to content assimilation among participants. These materials supported the development of accessible and context-sensitive pedagogical approaches that leveraged information and communication technologies (ICTs) to enhance the learning experience [38,39,40].

The development of the learning pathway “Food and Nutrition in ALS” on the AVASUS platform played a role in promoting knowledge translation and democratizing access to specialized content. While our findings indicate that the intervention supported the acquisition of evidence-based information relevant to ALS care, limitations such as low course completion rates and the voluntary nature of participation must be considered when interpreting the results. These findings highlight both the potential and the limitations of MOOCs as tools for continuing health education in complex and rare disease contexts.

Despite the advantages of MOOCs in expanding access to health education, their implementation and effectiveness present challenges that deserve critical reflection. High dropout rates, low completion rates, digital exclusion in underserved regions, and limited user feedback mechanisms are recurring concerns in the literature. Furthermore, assessing the long-term impact of such interventions remains methodologically complex. These limitations should inform future developments of open educational resources and guide strategies to improve user engagement and course effectiveness.

Reflecting on this educational intervention highlights its role in advancing both national and global agendas, notably the United Nations’ 2030 Agenda with its guiding principle “Leave no one behind” [41]. By addressing rare diseases like ALS, the learning pathway promotes equity and supports inclusive, sustainable development through health education, which intersects with multiple Sustainable Development Goals (SDGs).

This intervention directly contributes to SDG 2 (Zero Hunger) by providing knowledge to improve dietary and nutritional care for ALS patients, encompassing physiological, psychological, and sociocultural dimensions of nutrition. It aligns with SDG 3 (Good Health and Well-Being) by extending training beyond formal health professionals to family members and informal caregivers, thereby strengthening healthcare delivery and healthcare systems. In relation to SDG 4 (Quality Education), the pathway fosters relevant skills not only among health workers but also among patient families, representing an inclusive educational approach that drives social change [42,43,44].

Additionally, the strong female participation underscores its contribution to SDG 5 (Gender Equality), particularly through increased use of information and communication technologies to empower women. Finally, by offering free, scalable MOOCs to diverse audiences, including young people and adults, the pathway supports SDG 8 (Decent Work and Economic Growth) by promoting social inclusion, workforce development, and national progress [45,46].

Despite the significant contributions and impacts observed in this study, it is important to acknowledge certain limitations. As the intervention was delivered online and in a self-instructional format, participants may have faced challenges in accessing and completing the learning pathway, engaging in the study, completing all four courses, and participating in the evaluations. Additionally, the rarity and progressive nature of ALS posed further obstacles, as some participants with the disease may have been unable to complete all stages of the study. In light of these limitations, we recommend that future research adopt complementary approaches, combining quantitative and qualitative methodologies, to assess the effects of the educational intervention more comprehensively. Strategies such as personalized follow-up, interactive support, and enhanced accessibility features may help improve participation and learner engagement.

While this study provides valuable insights into the effectiveness of digital education in ALS nutrition, certain limitations must be acknowledged. First, the self-selection of participants may have introduced sampling bias, favoring individuals already motivated or familiar with the topic. Second, the reliance on digital access may have excluded participants from lower-resource settings or those with limited digital literacy. Third, as a before-and-after study without a control group, causality cannot be inferred. Additionally, the study did not assess course satisfaction, long-term retention, or real-world application of acquired knowledge. Future research should consider mixed-method designs to capture broader dimensions of learning and course impact.

## 5. Conclusions

This study demonstrated that a self-instructional, technology-mediated learning pathway can be an effective tool for disseminating knowledge about food and nutrition in the context of ALS. The intervention supported participants’ learning across all five regions of Brazil, highlighting the feasibility and reach of digital educational strategies for rare disease care.

At the regional level, the pathway expanded access to specialized training among healthcare professionals and caregivers in underserved areas, addressing educational gaps and promoting equity. At the global level, our findings reinforce the growing evidence on the effectiveness of MOOCs in health education—particularly in promoting scalable, inclusive learning opportunities for diverse audiences.

By combining open educational resources with flexible, learner-centered methodologies, this initiative contributed to more informed and competent care for individuals living with ALS. The model presented here may serve as a replicable framework for similar interventions focused on chronic and neurodegenerative conditions worldwide.

Continued investment in digital health education is recommended, along with strategies to increase course completion, enhance engagement, and ensure the inclusion of patients and caregivers in future programs. Future studies should explore knowledge retention, behavior change, and the impact of such interventions on clinical outcomes, especially in multidisciplinary care settings.

## Figures and Tables

**Figure 1 nutrients-17-02562-f001:**
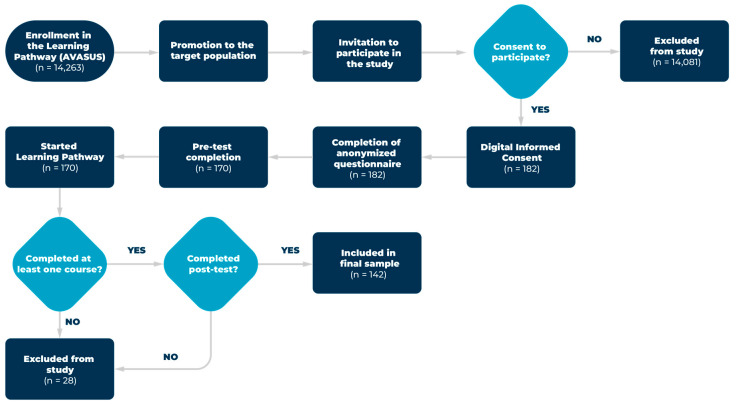
Flowchart illustrating the inclusion and exclusion process of participants in the study. The figure details the steps from course promotion and consent to pre- and post-testing, highlighting criteria for participant retention and exclusion.

**Figure 2 nutrients-17-02562-f002:**
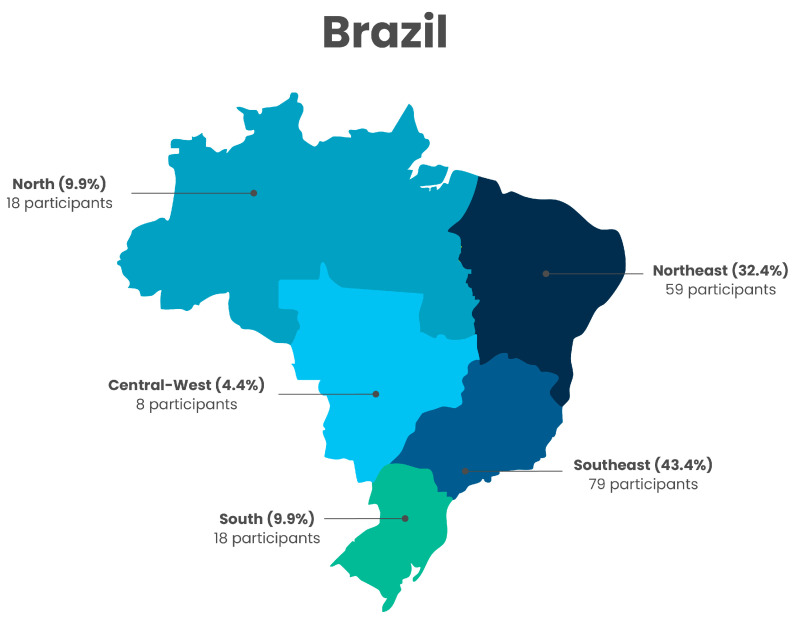
Geographic distribution of study participants across Brazil’s five macro-regions.

**Figure 3 nutrients-17-02562-f003:**
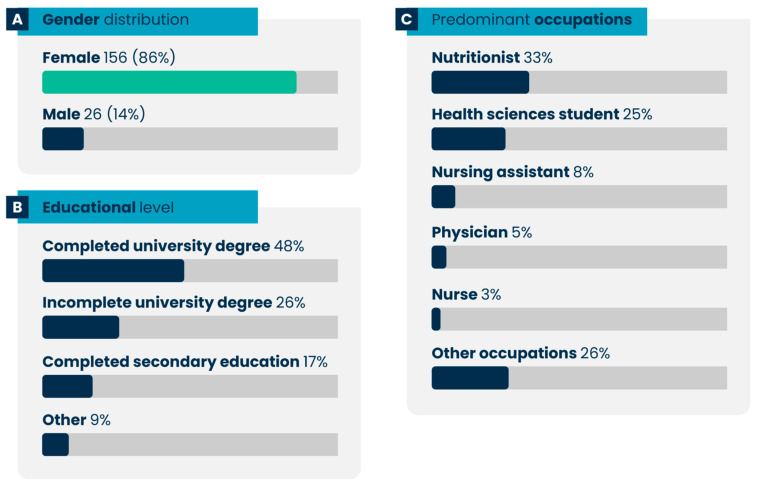
Sociodemographic and professional characteristics of study participants: distribution by (**A**) gender, (**B**) educational level, and (**C**) predominant occupations. Incomplete degree refers to participants who had started but had not yet completed a college degree.

**Table 1 nutrients-17-02562-t001:** Pre- and post-test scores of participants in each course of the learning pathway.

Learning Pathway Courses	Median (Q1–Q3)	*p **
1. Food and Nutrition for ALS Patients (n = 74)		<0.001
Pre-test score	9.17 (8.33–10.00)	
Post-test score	10.00 (10.00–10.00)	
2. Dietary Modifications for ALS Patients (n = 56)		<0.001
Pre-test score	9.17 (7.50–10.00)	
Post-test score	10.00 (10.00–10.00)	
3. Tube Feeding for ALS Patients (n = 72)		<0.001
Pre-test score	8.18 (6.36–9.09)	
Post-test score	10.00 (8.18–10.00)	
4. Specific Nutritional Considerations for ALS Patients (n = 59)		<0.001
Pre-test score	8.83 (6.67–9.17)	
Post-test score	10.00 (9.17–10.00)	

* Wilcoxon Test. Results are shown in median and interquartile range (Q1–Q3).

## Data Availability

The data presented in this study are contained within the article and available at Zenodo (https://doi.org/10.5281/zenodo.15678383 accessed on 21 July 2025).

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
