# Peer review of "Effectiveness of a Learning Pathway on Food and Nutrition in Amyotrophic Lateral Sclerosis"

_nutrients, 2025, doi:10.3390/nu17152562_

Round 1
Reviewer 1 Report
Comments and Suggestions for Authors
Dear colleagues,
Thank you for the opportunity to review the article "Effectiveness of a Learning Pathway on Food and Nutrition in Amyotrophic Lateral Sclerosis."
The first comment is that it is an interesting topic that captures the reader's attention.
However, for the article to be considered, all sections need significant revision.
Below you will find my detailed comments:
Title
Ok
Abstract
I will not comment at this point on the abstract, as it should be adapted according to the changes made to the other sections based on the reviewers' comments.
Keywords
The authors could define some of the key words more precisely, for example:
Why do the authors use motor neuron disease as a keyword, which is technically correct, when the article is primarily about amyotrophic lateral sclerosis?
Why is the term nutritional science used when the simpler and more contextually appropriate term is nutrition?
Why do you use the term health literacy when it is about the education of healthcare professionals, patients, and carers?
Please change or explain
Introduction
Lines 43–76: First, a general remark on the entire introduction. The authors dedicate the first part of the introduction to the importance of education by presenting AVASUS and MOOCs as educational models. I can only praise these educational approaches. However, in this introductory section, their role is presented in a consistently positive tone, without addressing potential drawbacks such as inequality in digital access, especially between urban and rural areas, the often low participation rates, and high drop-out rates in MOOC programmes. Neither the evaluations of these programmes nor the limitations of these learning formats are mentioned.
It would be beneficial if the authors also critically reflected on this aspect of educational approaches. As it stands, the introduction seems more promotional than scientific. A more balanced, scientific introduction would increase the credibility of the educational perspective presented.
Lines 77–98: The authors do not address the core topic and focus of the research, people affected by ALS as a significant public health issue, until this section. I suggest that the authors harmonise the entire introduction and further clarify the importance of education in the treatment of ALS, focusing on the nutritional aspect.
They should also address how a MOOC can effectively educate caregivers who do not have a medical background.
Materials and Methods
2.1. Ethical and legal aspects
Lines 105–107: “The study was approved..." – I kindly ask the authors to indicate the date on which the study was authorised.
Furthermore, how was the protection of sensitive data ensured, as a digital platform was used?
2.2 Study design and population
I recommend that the authors include a flowchart in this section to illustrate the process of inclusion and exclusion of participants in the study.
Lines 124–125: “The educational intervention..." – I kindly ask the authors to make this part more detailed and descriptive.
Lines 124–130: Were standardised and validated instruments used for the tests? Were they developed specifically for this study? What scoring system was used to assess the participants' knowledge? How many questions did the tests contain? What criteria were used to define success in the tests before and after the intervention? A more detailed description is needed
2.3. Data collection
Lines 136–138: What does “objective questions” mean? No examples are given for the questions. Please explain.
It would be helpful to describe the structure of the questionnaire and the type of questions and answer format — for example, multiple choice questions, Likert scale, etc.
If the questionnaire is not too long, it could also be included as a supplement to the article.
2.5. Data analysis
Lines 162–163: I kindly ask the authors to give precise details of the version and the publisher, taking copyright protection into account.
For example: IBM SPSS Statistics for Windows, Version 25.0 (IBM Corp., Armonk, NY, USA).
Results
Lines 169–170: “The ‘Food and Nutrition in ALS’ learning pathway…” — This entire paragraph belongs in the methodology section, where the pedagogical measures should be described.
Lines 174–177: “A total of 14,263..." — The authors state that 182 participants were included in the study, but only 142 completed at least one course, which was the inclusion criterion.
This means that out of a total of 14,263 potential participants, only 182 (1.3%) were enrolled in the study, and only 142 (0.99%) completed at least one course.
Such a significant drop in participation certainly requires further explanation.
Figure 1: Although the figure looks visually appealing, it seems counterproductive in the context of this article, as it is a very large country where only 8 participants from one region took part in the study.
Considering the large dispersion of an already small number of participants, this presentation further diminishes the value of the study.
It would therefore be more appropriate to omit this figure, especially as the authors do not comment at all on the significance of this data. Otherwise, I advise the authors to discuss this regional variability.
3.1. Characterization of participants
This is socio-demographic data, so the subtitle of this section is too general and unclear. I suggest that the authors clarify the title of the section and Figure 2.
As for the figure itself, the clear predominance of women among the participants is something that should be addressed both in the results and later in the discussion. However, the authors only present this as a fact without discussing or providing a possible explanation. It would certainly be useful to reflect on the gender distribution of ALS and the gender data obtained from the participants.
With regard to the level of education in Figure 2, it would be useful to carry out an additional statistical analysis to assess the statistical significance. Does the level of education influence participation in the educational programmes and the results achieved?
Also, the term “incomplete degree” should be explained in more detail. I recommend that the authors focus on the level of education completed rather than what participants started but did not complete.
Regarding the category “predominant occupations,” I remind the authors that they state in lines 111–113: "The target population included health professionals, formal and informal caregivers, and ALS patients of both sexes, from across Brazil."
However, this categorisation does not appear anywhere in the results, and it would be interesting to see differences between these groups. Instead, the authors list the occupations of the participants. Patients and informal carers may be among them, but this is not clear from the table.
Finally, I draw the authors’ attention to the fact that all these data, the categories mentioned in the results, need to be accurately described in the methodology of the study.
3.2. Educational Intervention of the Learning Pathway
Lines 200-201: “Among the 182 participants surveyed, 142 (78%) completed …” — Here we come back to the issue of inclusion and exclusion of participants in the study, and the flowchart suggested earlier in the methodology section would certainly help to clarify this.
It is needless to reiterate that 182 participants were surveyed, when only 142 completed at least one course. The results should relate specifically to these 142 participants.
Table 1: “This is a table. Tables should be placed in the main text near the first time they are cited”. I draw the authors’ attention to the fact that this cannot be the title of the table. The table shows the results of the training courses held.
Looking at the number of participants, it is clear that only a very small number took part in some of the educational modules. Certainly, there are differences in the results achieved, but it is not clear on what basis the scores were calculated. This needs to be detailed in the methodology section.
Lines 212–220: When discussing the modules within each course, these figures alone do not provide much insight, as the course details are not provided in the methodology, which is essential for further interpretation of the success, diversity, and themes of these courses.
Looking at the results, the question arises as to what extent the educational intervention contributes to increasing health literacy or specific knowledge about nutrition in ALS patients, which is supposedly one of the focal points of the research.
Discussion
Certainly, the introduction of new modern technologies should be supported, but as in the introduction, the authors adopt an almost promotional tone in the discussion with regard to AVASUS and MOOCs. What is missing is a critical reflection on the problems associated with the implementation, evaluation, and effectiveness of such methods.
Lines 254–256:
The authors note that most of the participants are women, but they merely state this as a fact without going into possible reasons for this gender distribution.
At the same time, they mention that the course participants were mostly highly educated people, especially nutritionists by profession. But what influence does the level of education have on the success of the training programme? I remind the authors of the title of the article: “Effectiveness of a learning pathway on food and Nutrition in amyotrophic lateral sclerosis”
The same question applies to the occupations of the participants. Where are the informal carers and patients? Although informal carers are mentioned, the specificities of this group are neither described nor further analysed in the results or discussion, although their important role is already emphasised in the introduction (lines 91–98).
What role could prior knowledge and motivation play in participation in the courses? How should the quality and content of the courses be assessed?
Finally, the discussion lacks a section in which the limitations of the study are explicitly pointed out.
Conclusions
I advise the authors to shorten the conclusions and make them more precise and concrete by clearly defining what new contributions this study has made at both the regional and global levels. In the first two paragraphs, this is initially addressed in part, but very quickly gets lost in general theoretical considerations.
Comments on the Quality of English Language
The article is linguistically correct. However, I advise a language check after revising the article according to the reviewers' instructions.
Author Response
Dear Reviewer,
Thank you for your thorough and constructive comments. We appreciate your attention to important aspects of our study. All the points raised have been carefully addressed through point-by-point responses to each reviewer comment. These responses are detailed in the PDF rebuttal letter attached to the submission, organized as follows:
- Reviewer 1 – pages 2 to 13
- Reviewer 2 – pages 14 to 16
- Reviewer 3 – page 17
- Reviewer 4 – pages 18 to 20
Additionally, with the extensive changes made in the updated version of the manuscript, we also revised the reference list. We made an effort to maintain the most essential and relevant citations while reducing self-citations and avoiding redundancy, thereby improving the scientific integrity and balance of the manuscript.
We believe that the revisions made, along with the clarifications provided in the rebuttal, have strengthened the transparency, rigor, and overall quality of our study. Thank you again for your valuable feedback.
Best Regards,
Lucia Leite-Lais

Reviewer 2 Report
Comments and Suggestions for Authors
This is a paper on an interesting initiative. Unfortunately, there are many issues which make it unsuitable for publication in its present state.
- The participation rate (182 out of 14,263) seems very low. Is there an explanation for this?
- Moreover, only 142 completed at least one course (l. 200), which contradicts the inclusion criteria (l. 113).
- ll. 126-130. "a post-test was aligned with the course content". This seems to imply that the post-test was different from the pre-test. This would preclude comparison. Please clarify.
- Fig 1: Why is the distribution over health professionals, caregivers and ALS patients (which is probably the most important) not shown?
- Table 1 and the text about this. Was 10 the maximum score? In that case the post-test for courses 1 and 2 seems very undiscriminatory: everyone had the maximum score. Also, at last 25% had already this score at the start, so did they learn anything new?
- In general, the empirical analysis is very rudimentary. The focus should be on the individual difference between pre-test and post-test. Account should be taken on how many courses people followed, and who they were (professionals ...). It should become clearer who took which course, with what results.
- The conclusion, and also large parts of the discussion are about the merits of AVASUS. I do not dispute those, but they are not outcomes of this study.
Author Response

(The authors gave the same response as above.)

Reviewer 3 Report
Comments and Suggestions for Authors
The atuhors presented a longitudinal qualitative study about the role of health education - by digital instuments - in patients with Amyotrophic Lateral Sclerosis. In order to improve the psychological health and pathology management of persons with Amyotrophic Lateral Sclerosis, health education is essential. Health education can substantially improve the quality of life and mental wellness of patients and caregivers by giving them the necessary information and coping mechanisms. In particular, about nutrition the authors should consider in future, further approach including multidisciplinary care (http://doi.org/10.1111/ene.14393; http://doi.org/10.1097/WCO.0b013e328356d328) in those patients including a precision medicine approach about metabolic and nutritional status. The role of medications on body composition (10.1007/s11357-010-9196-y; 10.1093/ajcn/nqac016) should be discussed.
Thank You
Author Response

(The authors gave the same response as above.)

Reviewer 4 Report
Comments and Suggestions for Authors
The manuscript submitted by Coutinho et al. to Nutrients has to be properly revised before it can be considered for publication in Nutrients. These are my suggestions:
The main results of the study must be expressed quantitatively in the abstract. Some directions for further investigations should also be indicated.
The Introduction is adequate and provides the needed background to understand the developed investigation.
The ethical approval date has to be declared.
I suggest the inclusion of a flowchart in section 2, with all the steps taken in this study for better comprehension.
The sample size seems not adequate; can you please justify it? This is clear when we see the Brazil map (Figure 1), and the participants are not representative of the study population.
The study limitations need to be better discussed in section 4.
The Conclusions should be more concise and be clear in the recommendation of directions for further investigations.
Author Response

(The authors gave the same response as above.)

Round 2
Reviewer 1 Report
Comments and Suggestions for Authors
Dear colleagues,
Thank you for the opportunity to review the article in its revised form. I would like to express my appreciation to the authors for the extensive changes they have made. In my opinion, these revisions have further improved the quality and appeal of the article to potential readers.
I would also like to thank the authors for their responses to the reviewers’ comments. The responses are appropriate and accurate. I congratulate the authors on a job well done. If possible, it would be helpful to include in Figure 1 the exact numbers that illustrate the inclusion and exclusion process of the study participants.
I recommend the article for publication in the journal and wish the authors every success in their future professional endeavours.
Reviewer 2 Report
Comments and Suggestions for Authors
Thank you for this very extensive revision of the article, which has clarified many aspects that were not clear in the previous version.